# Cleavage of the Perlecan-Semaphorin 3A-Plexin A1-Neuropilin-1 (PSPN) Complex by Matrix Metalloproteinase 7/Matrilysin Triggers Prostate Cancer Cell Dyscohesion and Migration

**DOI:** 10.3390/ijms22063218

**Published:** 2021-03-22

**Authors:** Tristen V. Tellman, Lissette A. Cruz, Brian J. Grindel, Mary C. Farach-Carson

**Affiliations:** 1Department of Diagnostic and Biomedical Sciences, School of Dentistry, The University of Texas Health Science Center at Houston, 7500 Cambridge Street Room 4401, Houston, TX 77054, USA; Tristen.Tellman@uth.tmc.edu (T.V.T.); Lissette.Cruz@uth.tmc.edu (L.A.C.); BJGrindel@mdanderson.org (B.J.G.); 2Department of BioSciences, Weiss School of Natural Sciences, Rice University, Houston, TX 77005, USA; 3Center for Theoretical Biological Physics, Rice University, Houston, TX 77005, USA

**Keywords:** matrilysin/MMP-7, perlecan/HSPG2, microtumors, dyscohesion, prostate cancer, migration

## Abstract

The Perlecan-Semaphorin 3A-Plexin A1-Neuropilin-1 (PSPN) Complex at the cell surface of prostate cancer (PCa) cells influences cell–cell cohesion and dyscohesion. We investigated matrix metalloproteinase-7/matrilysin (MMP-7)’s ability to digest components of the PSPN Complex in bone metastatic PCa cells using *in silico* analyses and in vitro experiments. Results demonstrated that in addition to the heparan sulfate proteoglycan, perlecan, all components of the PSPN Complex were degraded by MMP-7. To investigate the functional consequences of PSPN Complex cleavage, we developed a preformed microtumor model to examine initiation of cell dispersion after MMP-7 digestion. We found that while perlecan fully decorated with glycosaminoglycan limited dispersion of PCa microtumors, MMP-7 initiated rapid dyscohesion and migration even with perlecan present. Additionally, we found that a bioactive peptide (PLN4) found in perlecan domain IV in a region subject to digestion by MMP-7 further enhanced cell dispersion along with MMP-7. We found that digestion of the PSPN Complex with MMP-7 destabilized cell–cell junctions in microtumors evidenced by loss of co-registration of E-cadherin and F-actin. We conclude that MMP-7 plays a key functional role in PCa cell transition from a cohesive, indolent phenotype to a dyscohesive, migratory phenotype favoring production of circulating tumor cells and metastasis to bone.

## 1. Introduction

Among the non-collagenous components of the extracellular matrix, the large heparan sulfate proteoglycan perlecan/heparan sulfate proteoglycan 2 (HSPG2) plays a unique role in coordination of signaling complexes on the surfaces of the cells that it engages. As we previously proposed [1], perlecan can be considered as an extracellular scaffolding protein in which each of its five structural domains interacts with various classes of surface complexes including growth factor receptors (domain I), low-density lipoproteins (domain II), calcium channels (domain III), cell–cell adhesion (domain IV) and cell–substratum interactions (domain V) [2,3]. Our laboratory has spent many years exploring the detailed functions of domain IV in the human protein, which consists of 21 Ig-like modules in three functional subdomains (1–3) with two basic structural motif types [2]. Among these, perlecan domain IV-3 (Dm IV-3), the last seven Ig repeats, interacts with semaphorin 3A (Sema3A) in complex with cell-surface plexin A1 and neuropilin-1 (NRP1) to regulate epithelial cell adhesion and dyscohesion [4]. Control of this clustering behavior by Dm IV-3 has been observed in both cancer [5] and normal [6,7] cells. Plexins and semaphorins influence cell migration through integrin activation and cytoskeletal reorganization [8] and have differential expression levels relative to prostate cancer (PCa) disease state, demonstrating their importance in PCa progression [9]. Previously published work in our lab established a novel interaction between Dm IV-3 and Sema3A [4]. Figure 1 provides a schematic of the Perlecan-Semaphorin 3A-Plexin A1-Neuropilin-1 (PSPN) Complex. This complex forms as a dimer of heterotrimers stabilized by perlecan as demonstrated in the figure [10]. It is important to note that the fifth β-propeller blade of Sema3A interacts at the a1 domain of NRP1 that subsequently interacts with the sixth β-propeller blade of plexin A1 [10].

Perlecan, when fully decorated with its three to four glycosaminoglycan (GAG) chains consisting of a cell type-dependent ratio of heparan and chondroitin sulfate chains [11], is highly resistant to proteolysis by most extracellular proteases [12]. Removal of the GAG chains by endoglycosidases renders the core protein more susceptible to digestion [12]. One matrix metalloproteinase (MMP), MMP-7, possesses the ability to extensively digest the perlecan core protein while still fully decorated with GAG [5]. Digestion with MMP-7 releases the perlecan scaffolding function and alters intracellular signaling including from the PSPN Complex, a phenomenon we have referred to as a “molecular switch” for dyscohesion [4,5,13]. Cleavage of perlecan by MMP-7 occurs from the C-terminal, releasing protein fragments of various sizes primarily from domains IV and V [5]. These fragments were mapped by mass spectrometry and shown to be present in the serum of patients with metastatic cancer [14]. Among these, one domain IV-derived MMP-7-produced fragment contains a 17 amino acid peptide sequence, perlecan 4 (PLN4) peptide, that we had previously identified to have a unique activity in cell–substratum adhesion and activation of focal adhesion kinase (FAK) [15]. Previous work also showed that full-length, fully decorated perlecan (FL pln) induces clustering of PCa cells when pre-coated onto surfaces [4,5].

Cadherins are primary mediators of cell–cell adhesion. E-cadherin performs this dynamic function in epithelial cells particularly for homotypic interactions among like-type cells [16]. Stabilization of E-cadherin-mediated binding is supported by cortical actin that “locks” cell–cell adhesion; thus, co-localization of E-cadherin and the actin cytoskeleton can serve as an index of tight cell cohesion [17]. MMP-7 can cleave E-cadherin, leading to shedding of the ectodomain and cell dyscohesion [18,19]. The ability of MMP-7 to cleave the entire PSPN Complex, further triggering dyscohesion and cell migration, has not been previously explored.

In this work, we assessed the ability of MMP-7 to digest all components of the PSPN Complex in the context of PCa, a metastatic disease with proclivity to metastasize to the perlecan-rich bone marrow where it further metastasizes by production of circulating tumor cells (CTCs) [2,20,21]. Using a preformed microtumor model developed for this project, we also examined the initiation of cell migration after MMP-7 digestion and tested the hypothesis that the bioactive PLN4 peptide produced by MMP-7 digestion of perlecan could further enhance cell migration. Finally, we investigated the ability of digestion with MMP-7 to destabilize cell–cell junctions in microtumors using co-registration of E-cadherin and F-actin as a quantitative tool to study the quality of cell adhesion complexes.

## 2. Results

### 2.1. In Silico Digestions of Plexin A1 and Neuropilin-1 by MMP-7

*In silico* digestion of FL pln [5] and Sema3A [4] was reported previously. SitePrediction online software was used to analyze the ability of MMP-7 to cleave cell-surface plexin A1 and NRP1 *in silico*. Schematics of plexin A1 and NRP1 (Figure 2A,B) show the top ten predicted MMP-7 cleavage sites, all with specificity >99%. Predicted cleavage sites are indicated on the schematic with the cleavage sequence and accompanying rank order, with 1 representing the most likely cleavage. The plexin A1 most likely cleavage site produced a 137.5 kilodalton (kDa) N-terminal fragment that contained the interacting sema domain (Figure 2A). The most likely NRP1 cleavage site in the VEGF b1 domain produced a 64.8 kDa N-terminal fragment containing the Sema3A-plexin A1-binding a1 domain (Figure 2B). Predicted ribbon structures were created *in silico* to demonstrate the availability of predicted cleavage sites within the context of three-dimensional space. Cleavage sites are indicated by color-coded keys (Figure 2C,D). PHYRE modeling could not accurately predict all regions of the plexin A1 and NRP1 structures because of a lack of available homology, so not all cleavage sites could be represented. *In silico* predictions thus demonstrated the likelihood that MMP-7 could cleave both NRP1 and plexin A1 at multiple cleavage sites (Figure 2).

### 2.2. Demonstration of Digestion of All PSPN Complex Components by MMP-7

Previous work from our lab demonstrated the ability of MMP-7 to cleave FL pln with intact heparan sulfate chains [5]. Dm IV-3 was cleaved into limit peptides with proposed bioactivity [5,15]. Purified Dm IV-3 was digested with MMP-7 into these limit peptides, consistent with FL pln digestions, with five major bands containing these fragments [5]. The resulting Dm IV-3 fragments ranged from 17 to 53 kDa. Recombinant Sema3A, a secreted member of the PSPN Complex, was digested with MMP-7 to assay for cleavage sequences within the protein then electrophoresed using SDS-PAGE and visualized by silver staining. Digestions yielded multiple bands, the most prominent of which was approximately 30 kDa [4]. A recombinant portion of plexin A1 ectodomain was incubated overnight with MMP-7 to validate *in silico* predictions. The resulting silver stain showed extensive proteolysis, producing a single major band at 15 kDa, indicative of multiple cleavages within the predicted regions (Figure 3C). Following *in silico* predictions, recombinant NRP1 was digested overnight with MMP-7 to determine experimentally the ability of MMP-7 to cleave NRP1 at the predicted cleavage sequences. A silver stain of the digestion showed nearly complete digestion of the protein, with two major bands detected at 20 and 15 kDa. Taken together, these results indicated that MMP-7 plays a major role in the cleavage and resulting destabilization of the PSPN Complex.

### 2.3. MMP-7 Promotes and Perlecan Inhibits Cell Dispersion

To determine whether preformed PCa microtumors will retain their clustered, cohesive phenotype when exposed to soluble perlecan, MMP-7 or PLN4 peptide [15], the microtumor formation assay was developed (Figure 4). A 24-well plate microwell system was used to form similarly sized microtumors (Figure 4(A″,A‴)). C4-2 PCa cells were chosen among several PCa cell lines we tested (Appendix A), because they responded to perlecan by clustering and did not produce detectable levels of endogenous MMP-7 that prevents clustering. C4-2 cells were seeded at a density of 100,000 cells per well and incubated for one day (Figure 4(A′)). On day 2, microtumors were transferred to FL pln (pre-treatment) and incubated for 24 h (Figure 4B). On day 3, microtumors were transferred to 96-well plates and a treatment was added to the cell culture media—bovine serum albumin (BSA), FL pln, MMP-7, PLN4 peptide or PLN4 peptide plus MMP-7—and the behavior of cells was followed by continuous live-cell imaging (Figure 4C). On day 4, a second, identical treatment was performed (Figure 4C). On day 5, live-cell imaging was stopped, and images were analyzed (Figure 4D).

Figure 5 shows the results of these experiments, examined for differences in treatment and time. The spreading area of microtumors treated with FL pln was half in comparison to those treated with MMP-7 at all time points (Figure 5B). Note that the mean fold change area values inside the red squares (Figure 5C) at 48 h FL pln highlight the fold change value is similar to MMP-7 and MMP-7 plus PLN4 peptide values. This suggests that microtumors continuously treated with FL pln display a suppression in cell dispersion area and that this persisted at all time points (Figure 5C, Appendix A) Microtumors treated with MMP-7 rapidly dispersed to create large irregular structures with exposed regions of the plate (Figure 5A). PLN4 peptide alone did not have a significant effect on the cell dispersion area (Figure 5B), but clusters treated with the PLN4 peptide plus MMP-7 tended to disperse farther and more irregularly than those with MMP-7 alone. (Figure 5A,B). In the PLN4 peptide plus MMP-7 treated cells, rapidly spreading cells leaving dyscohesive microtumors also left empty spaces (Figure 5A). The difference in cell dispersion area became striking at the 36 h time point for MMP-7 and MMP-7 plus PLN4 peptide-treated microtumors; we observed a ~6-fold and ~7.6-fold increase in area, respectively (Figure 5B,C). Overall, a near-complete dispersion was observed in all treatment groups except for microtumors treated with FL pln at 48 h. For all treatments, most cells migrated out collectively from the microtumors, and remained in contact with other cells. Some complete cell detachment was observed, with some cells migrating in small groups or as single cells in all treatments. These data show that MMP-7 can disperse microtumors pre-treated with FL pln, while the continuous presence of FL pln impairs cell spreading and favors a cohesive phenotype. The presence of MMP-7 and a FAK-activating peptide derived from perlecan is the most dyscohesive treatment that we tested (Figure 5C).

### 2.4. MMP-7 Disrupts Cadherin-Based Cohesion

To assess cell adhesion complexes during microtumor cohesion/dyscohesion, microtumors were pre-treated with Dm IV-3 on day 2 (Figure 4B) and transferred to collagen I-coated wells on day 3 (Figure 4C). E-cadherin and F-actin immunostaining were performed 24 h after treatment with Dm IV-3 and MMP-7 (Figure 6) and the extent of co-registration of signal used as an index of cell–cell adhesion. Fluorescence intensity profiles of E-cadherin and F-actin at the zones of cell–cell interactions (black arrows on linescans) showed high signal registration in Dm IV-3-treated microtumors at cell–cell contacts, while the co-aligned intensity peaks in microtumors treated with MMP-7 were substantially fewer. These findings demonstrate that MMP-7 treatment of PCa cells results in rearrangement of E-cadherin and F-actin at cell–cell contacts that can contribute to dyscohesion and cell dispersion from microtumors. Interestingly, the microtumors treated with MMP-7 showed more invadopodia than those left untreated. Microtumors treated with Dm IV-3 showed an increase in E-cadherin and F-actin at cell–cell contact zones at the center and at the periphery. Thus, the likelihood of PCa cells leaving microtumors treated with Dm IV-3 is decreased compared to those treated with MMP-7.

## 3. Discussion

The dynamic interaction between MMP-7 and perlecan in metastatic PCa cells is a key regulator of cell behavior in the tumor microenvironment [5,14]. Chronic inflammation of the tumor microenvironment surrounding PCa tumors increases perlecan production, creating a perlecan-rich stromal matrix [22]. For PCa cells to metastasize, they must degrade the perlecan-rich matrix and create space to move. To do this, they either need to produce a combination of proteases able to degrade the core protein along with GAGases able to first cleave away the GAG chains, or they can produce a metalloproteinase that is able to cleave fully decorated perlecan such as MMP-7. Highly aggressive PCa cells produce significant amounts of MMP-7 [14,23,24]. We showed previously that MMP-7 can cleave native perlecan bound up in the extracellular matrix, allowing PCa dispersion and invasion through the matrix [5].

Until recently, the mechanism of receptor activation by perlecan at the PCa cell surface remained elusive. This changed with the identification of a new surface complex, dubbed the PSPN Complex, which exists as a dimer of heterotrimers stabilized by perlecan via an interaction between Dm IV-3 and the Ig region of Sema3A, forming a homotypic Ig–Ig interaction, as shown in Figure 1. This dimerization is essential to the ability of the complex to activate downstream signaling as shown by several groups [25,26]. Based on crystal structures of the Sema3A–NRP1–plexin A1 interaction at the sema domains, the a1 region of NRP1 forms a stabilizing brace to facilitate interactions between Sema3A and plexin A1 [10]. Previous work identified the interaction between Dm IV-3, Sema3A, and MMP-7, but no studies investigated the impact of activated MMP-7 on other PSPN Complex components. Initially, we looked to predictive models to provide insight into PSPN Complex dynamics and found numerous cleavage sites within both plexin A1 and NRP1. Subsequent in vitro digestions demonstrated that MMP-7 proteolysis can destroy all components of the PSPN Complex, an event that we showed favors dyscohesion, dispersion and cell migration, events associated with formation of CTCs and metastasis in PCa. Limit fragments resulting from the digestion of Dm IV-3 by MMP-7 were analyzed previously by our group where we predicted these to have bioactive roles to enhance PCa cell dispersion [5]. It is not unreasonable then to assume that derivative protein fragments resulting from PSPN Complex cleavage could perpetuate PCa dispersion and metastasis given the role of the SPN portion of this complex identified in cell migration. It is worth noting here that the region of NRP1 that forms the bridge between Sema3A and plexin A1 is a likely portion to be released by MMP-7 cleavage of the protein.

To investigate the impact of perlecan and MMP-7 during microtumor cell dispersion, we developed the microtumor formation assay. The bone metastatic PCa C4-2 cells used in this study are an androgen-insensitive cell line derived from LNCaP cells with little inherent MMP-7 production in the absence of inflammatory cytokines (Appendix A), making them an ideal candidate for this type of analysis in investigating MMP-7′s role in dispersion [27,28]. We note that among other cell lines of different prostate cancer subtypes (PCS) we examined, LNCaP cells (PCS2) behaved similarly to C4-2 in clustering assays and expressed low amounts of MMP-7 whereas two other highly aggressive cell lines, LNCaP RANKL (PCS not profiled) and PC3 (PCS3), both of which constitutively produce MMP-7, either failed to cluster or formed small clusters (Appendix A) [29]. We have not systematically measured dispersion, E-cadherin or F-actin rearrangements in the LNCaP line. In this paper, we report our findings that FL pln decreased cell dispersion from the C4-2 microtumors, whereas MMP-7 led to a complete cell dispersion phenotype. We also present a qualitive determination of cell–cell adhesion states, where microtumors treated with Dm IV-3 present more co-aligned E-cadherin and F-actin at cell–cell contacts than do microtumors treated with MMP-7. Several reports showed that both E-cadherin and F-actin need to be present at cell–cell contact zones to maintain close cohesion between cells [30,31,32]. As a result, the decrease in E-cadherin and F-actin at cell–cell contacts zones on microtumors treated with MMP-7 can result in microtumor dyscohesion, such as would occur when tumors release CTCs.

Prostate cancer is not a homogeneous disease. It was of interest that the levels of MMP-7 produced by the various PCa cell lines that we tested, representing different PCS, directly correlated with their ability to cluster. Both LNCaP and C4-2 cells are of a more indolent PCS2 as recently profiled by comprehensive study of gene expression [9] and made little or no MMP-7. In contrast, a highly metastatic, bone destructive subline of LNCaP expressing RANKL and in which c-Myc is active [29] did not cluster suggesting these cells are already dyscohesive. PC3 cells, representing another aggressive PCa subtype PCS 3 [9] fell somewhere in the middle, displaying some clustering that fell short of that observed with the LNCaP series lines. Previous work in our lab [4] determined that the AKT pathway lies downstream of the PSPN Complex (Figure 6) and is activated by MMP-7. Pathway analysis of the Prostate Cancer Transcriptome Atlas (PCTA) [9] showed that the AKT pathway is most important in cells and tumors with the PCS2 subtype (Appendix A). We confirmed by western blot with two commonly used prostate cancer cell lines of different subtypes that active pAKT levels were higher in PCS2 line C4-2 than in PCS3 line PC3 (Appendix A). We suggest that our findings reported here are most applicable to PCa tumors with a luminal PCS2 subtype, and perhaps a more aggressive luminal PCS1 subtype, and less relevant for cancers that already produce significant amounts of MMP-7 that can readily cleave the PSPN Complex components.

The release of pro-migratory bioactive fragments from some or all members of the proteolyzed PSPN Complex could further increase dispersion of intact PCa cell clusters. PLN4 peptide did not itself increase microtumor cell dispersion, but when combined with MMP-7, cell dispersion area increased noticeably over MMP-7 alone. This suggests that for PLN4 peptide to increase cell motility, the cell–cell adhesions must first be loosened by MMP-7. Future work will determine whether other perlecan fragments or other components of the PSPN Complex resulting from MMP-7 digestion have this same ability. The usefulness of the microtumor formation assay reported here is that it allows us to combine quantitative migration analyses and more detailed morphological and molecular analyses to fully determine the impact of PLN4 peptide or other bioactive fragments generated by MMP-7. In the context of understanding the mechanisms that underly MMP-7′s ability to facilitate metastasis of PCa, it is useful to link the events occurring at the cell surface with intracellular signaling pathways and gene expression. In previous analyses of differences in the phosphoproteomes in perlecan-induced cohesive and MMP-7-treated dyscohesive cells, we found that the PSPN Complex signaled to downstream FAK and FOXM1 [4]. Additional work showed the downstream influence of this molecular switch on transcription factor FOXM1, which binds to the MMP-7 promoter and increases transcription [33]. We propose that activation of MMP-7 in an inflammatory environment and resulting turnover of the PSPN Complex, the full proteolysis of which is described here for the first time, may activate a feed-forward loop, increasing localized MMP-7 production and perpetuating a highly dyscohesive PCa phenotype. One intriguing idea currently under study, supported by the aggressive metastatic behavior of the LNCaP-RANKL cells, is that c-MET, a direct upstream regulator of FOXM1 [34], is responsible for stimulating production of MMP-7 and favoring dyscohesion. Figure 7 provides a model of the transition in cell phenotype that we envision occurring as a consequence of PSPN cleavage by MMP-7. Notably, chronic activation of FOXM1 is a hallmark of metastatic cells [35], and FOXM1 was found to be highly active in aggressive but not indolent PCa cells including CTCs [9].

## 4. Materials and Methods

### 4.1. Cell Culture

PCa C4-2 cell lines were cultured in 10% (*v*/*v*) heat-inactivated fetal bovine serum (FBS) (Atlanta Biologicals S11150, Minneapolis, MN, USA) in RPMI 1640 (Gibco 22400105, Dublin, Ireland) with 1x penicillin/streptomycin (P/S) (Gibco 15140122). Cells were passaged at 80–90% confluency with 0.25% (*w/v*) trypsin EDTA (Gibco 25200072) with a 1:8 seeding density. HT-29 cells were cultured in 10% FBS and 1x P/S in DMEM (Gibco 11965118) for passaging and expansion. All cells were incubated at 37 °C in a humidified 5% (*v*/*v*) CO_2_ atmosphere. C4-2 cells were karyotyped and routinely monitored for mycoplasma infection and retention of PCa biomarkers. LNCaP, LNCaP RANKL and PC3 cells were cultured as described previously [5,29].

### 4.2. Perlecan Purification

Full-length perlecan was purified from the conditioned medium of HT-29 cells (formerly called WiDr) (ATCC CCL-218, Manassas, VA, USA) as described previously [5,36]. HT-29 cells were seeded in a hyperflask (Corning 10030, Corning, NY, USA) in 10% (*v/v)* FBS 1x P/S DMEM and switched to 2% (*v/v)* FBS 1x P/S DMEM at nearly 100% confluence. Full media exchanges were conducted every three days. Phenylmethylsulfonylfluoride (PSMF), benzamidine, and EDTA at 2.5 mM each and 0.02% (*w*/*v*) sodium azide were added to collected conditioned media and filtered through a 0.2 µm PES filter flask (Corning 431098). 500 mL of conditioned media from cells was concentrated using an Amicon Stirred Cell with a 100 kDa molecular weight cutoff filter (Millipore Sigma PBHK07610, Burlington, MA, USA). Concentrated conditioned medium was passed through a diethyl aminoethyl anion exchange (DEAE) column (Cytiva 17070910, Marlborough, MA, USA). Equilibration buffer contained 2 M urea, 50 mM HEPES, 250 mM NaCl, 2.5 mM EDTA, 0.5 mM benzamidine, 0.5 mM PMSF, and 0.02% sodium azide. Elution buffer was identical aside from 750 mM NaCl. A volume of 10 mL of elution buffer were added to the column and the elution was collected in 1.5 mL fractions then absorbance read at 280 nm. Fractions with high absorbance were pooled and buffer exchanged to MilliQ water using centrifugal filters (Millipore Sigma Amicon UFC905008). The buffer-exchanged samples were subjected to size-exclusion separation via Sepharose CL-4B (Millipore Sigma CL4B200) gel filtration chromatography. The 1.5 mL fractions from sample flow through were collected followed by 10 mL of 0.8 M NaCl. Remaining DEAE fractions and CL-4B fractions were analyzed by dot blot immunoassay with anti-perlecan Dm I antibody A71 (Invitrogen CSI 001-71-02, Carlsbad, CA, USA). Positive perlecan fractions were pooled and dialyzed in final PBS buffer (Millipore Sigma Amicon UFC905008). Samples were stored as aliquots at −80 °C. Perlecan Dm IV-3 was recombinantly produced in HEK293A cells as described previously [5].

### 4.3. In Silico Predictions

The amino acid sequences of plexin A1 (uniprot ID: Q9UIW2) and NRP1 (uniprot ID: O14786) were subjected to *in silico* digestions using the cleavage prediction site SitePrediction [37]. Predictions were generated using information for matrix metallopeptidase-7 (Homo sapiens, P4-P2′, 137) and default settings. The top 10 cleavage sequences are reported, all with a specificity score of >99%. Three-dimensional predictions of protein structures were modeled using PHYRE2 [38].

### 4.4. MMP-7 Digestions and Silver Stains

Recombinant human plexin A1 (Abcam ab226422, Cambridge, MA, USA) and recombinant human NRP1 (Novus Biologicals 3870-N1-025, Centennial, CO, USA) were subjected to digestion by recombinant human MMP-7 (Millipore CC1059). The digestion buffer contained 10 mM HEPES buffer pH 7.0, 3 mM calcium acetate, and 1 mM EDTA. Digestions for plexin A1 and NRP1 contained 2 µg purified protein and 0.2 µg MMP-7. Digestions were incubated at 37 °C for 16 h and heat denatured at 100 °C in 5x Laemmli buffer for 5 min. Denatured samples were loaded on Novex NuPAGE 10% Bis-Tris protein gels, 1.0 mm, 10 well (Invitrogen NP0301BOX,) and electrophoresed at 150 volts with 1x MOPS buffer (Invitrogen NP0001) for approximately 80 min. After electrophoresis, gels were stained with Thermo Scientific™ Pierce™ Silver Stain Kit (Thermo Scientific™ 24612, Waltham, MA, USA) according to manufacturer’s directions. Dm IV-3 and Sema3A digestions were conducted as described previously [4,5].

### 4.5. Microtumor Formation Assay

C4-2 cells were seeded at a density of 100,000 cells per well into a 24-well plate containing 750 round bottomed microwells per well (5D Spherical plate, Kugelmeiers, Zurich, Switzerland) and allowed to form microtumors following the manufacturer’s protocol. During microtumor formation, cells remained in RPMI without phenol red (Gibco 11835030) containing 1% (*v/v*) FBS, 1x P/S in a 95% humidity and 5% (*v/v*) CO_2_ incubator at 37 °C for 24 h. Microtumors were harvested and transferred to Dm IV-3 or FL pln-coated wells. After a 24 h incubation period, microtumors were transferred to uncoated or rat collagen I (R&D Systems 3440-005-01, Minneapolis, MN, USA)-coated 96-well plates and at this time FL pln (3 µg), Dm IV-3 (4 µg), PLN4 peptide (3 µg) and MMP-7 (0.04 µg) were added. The final volume was ~250 µL. PLN4 peptide was synthesized and used as described previously [15].

### 4.6. Live-Cell Imaging and Image Analysis

An A1R MP+ microscope (Ti Microscope) equipped with a Plan Apo λ 10X objective from Nikon Instruments was used for time lapse studies of dyscohesion and migration. Brightfield live-cell imaging was performed with a temperature-controlled heated stage at 37 °C over ~48 h and images were taken every 25 min. Images were imported into Volocity Software 6.0.1 (PerkinElmer Inc., Waltham, MA, USA) to calculate the area of cell dispersion from microtumors at the indicated time points. To represent the total microtumor dispersion area, the drawn perimeter consisted of spreading cells migrating out of the microtumors and free the spaces created by migratory cells at later time points. Briefly, regions of interest at the perimeter of spreading cells were defined using the Free Hand tool at the microtumor periphery (yellow dotted lines in Figure 5) during the initial time points; and as cell dispersion from microtumors occurred. The few cells that left microtumors as single cells were not included in the analysis. The area values (µm^2^) were exported in a comma separated value (.csv) format. Using Excel, the change fold in area was calculated by dividing spreading area at the indicated time points by the spreading area at time 0. Change fold in area values were analyzed using GraphPad Prism 9.0.1 software (San Diego, CA, USA).

### 4.7. Antibodies and Indirect Immunofluorescence

All steps were performed at room temperature unless otherwise specified. For analysis of cell cohesion and dyscohesion, microtumors were fixed with 4% (*w/v*) paraformaldehyde for 20 min and then permeabilized with 0.3% (*v/v*) Triton X-100 for 30 min with gentle agitation. Samples were blocked with 3% (*w/v*) BSA, 3% *(v/v)* goat serum, 0.3% Triton X-100 diluted in 1x PBS. Next, samples were incubated with rabbit anti-E-cadherin antibody (1:300, Signaling, 3195S) overnight at 4 °C. Samples were washed with 1x PBS six times and incubated with secondary antibody: 568 conjugated goat anti-rabbit (1:1000, Life Technologies, Carlsbad, CA, USA) and 488 conjugated phalloidin (1:100, Invitrogen) staining overnight at 4 °C in the dark. Samples were washed four times with 1x PBS and imaged.

### 4.8. Image Acquisition for Indirect Immunofluorescence and Line Scan Analysis

A1R MP+ microscope (Ti Microscope) equipped with Apo LWD 40x WI λS DIC N2 objective from Nikon was used to image immunostained microtumors. To assess the quality of adherens junction-based cell–cell adhesions line scan analysis on Figure 6 was done using Volocity Software 6.0.1 (PerkinElmer Inc., Waltham, MA, USA).

### 4.9. Enzyme-Linked Immunosorbent Assay (ELISA)

PCa cells were split and seeded at 150,000 cells per well in a 12-well plates in triplicate. Cells were seeded in 1 mL of normal culture media and cultured for 40 h. Conditioned media samples were assayed using the Human Total MMP-7 Quantikine Kit (R&D Systems DMP700) according to manufacturers’ instructions. Cells were cultured as described above except for the LNCaP RANKL cells which were cultured without G418 for this particular assay.

### 4.10. Statistical Analysis

For statistical analyses, GraphPad Prism 9.0.1 was used. Normalized cell dispersion areas are shown as the mean ± SEM, with three experimental repetitions except for PLN4 peptide plus MMP-7 which are two experimental repetitions. Two-way repeated-measures ANOVA was used to identify interaction between different treatments and time. These tests were followed by a *post hoc* Šídák’s multiple comparisons test. Results of the two-way repeated-measures ANOVA can be found in Appendix A.

## 5. Conclusions

In summary, this work demonstrates that a newly identified cell surface complex, the PSPN Complex, consisting of the extracellular matrix protein, perlecan, interacting with the cell surface complex Sema3A, plexin A1 and NRP1, plays a key functional role in cell cohesion and dyscohesion in metastatic PCa cells. Cleavage of all components of the PSPN Complex by MMP-7 is possible, and can activate signaling pathways that include FAK, AKT and FOXM1, decreasing cell cohesion and increasing cell motility and dispersion. A FAK-activating peptide from a region of perlecan cleaved by MMP-7 (PLN4) additionally augments motility in the presence of MMP-7. While MMP-7 inhibitors previously failed in clinical trials for treatment of metastatic disease [39,40,41], the findings reported here point to new avenues to interfere with downstream activation of dyscohesion and migration resulting from MMP-7 cleavage of the PSPN Complex. This opens the door to investigation of combination therapies with a new generation of specific MMP-7 inhibitors or inhibitors of the downstream pathways including FOXM1.

## Figures and Tables

**Figure 1 ijms-22-03218-f001:**
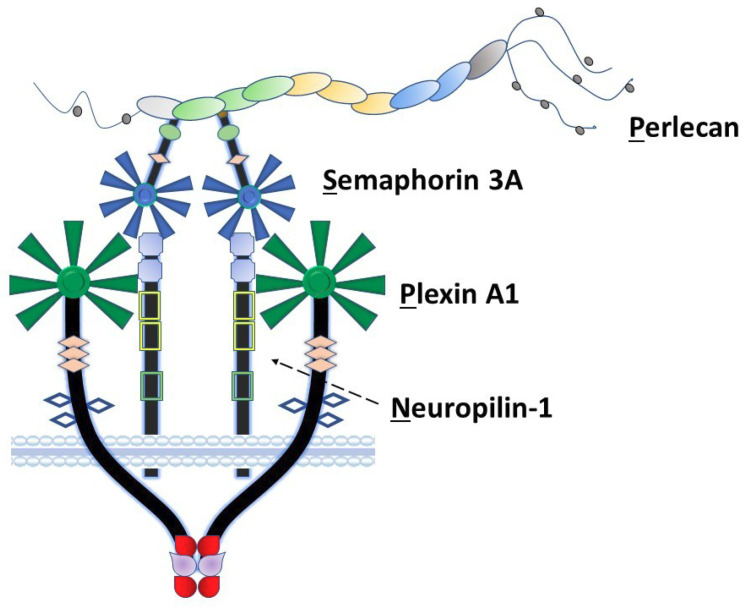
The PSPN Complex forms as a dimer of heterotrimers stabilized by extracellular perlecan. Perlecan/HSPG2 interacts with secreted Sema3A to bind NRP1 (dotted arrow) and plexin A1 to stabilize the PSPN Complex and prevent PCa dyscohesion. Dm IV-3 of perlecan (light green oval) interacts with the Ig module (light green) in the Sema3A dimer. Subsequently, this stabilized dimer interacts with the a1 domain of NRP1 (light blue square), which acts as a bridge between Sema3A and plexin A1. The sema domains of both Sema3A (blue) and plexin A1 (dark green) interact at this a1 NRP1 domain as a dimer of heterotrimers.

**Figure 2 ijms-22-03218-f002:**
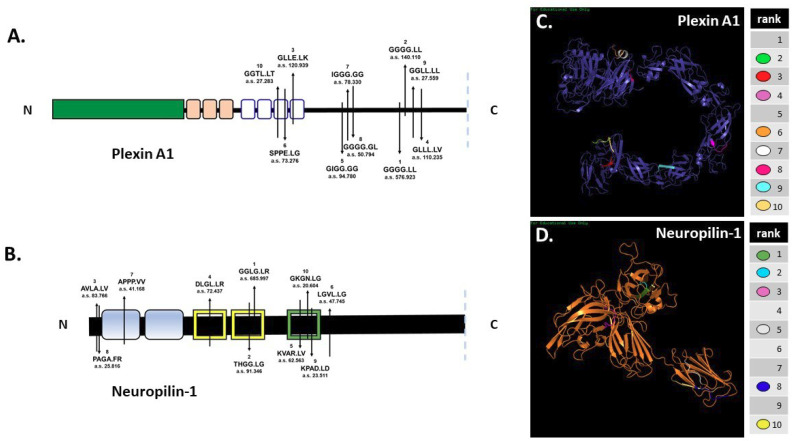
*In silico* predictions demonstrate MMP-7′s potential to cleave plexin A1 and NRP1. *In silico* digestion of (**A**) plexin A1 and (**B**) NRP1 using MMP-7 cleavage sequence information in SitePrediction online software. Schematics of plexin A1 and NRP1 are shown with representative domains and relative location of the top ten predicted cleavage sites. Numbers represent the ranked score for each sequence with the corresponding amino acid sequence, with exact cleavage indicated by the period. PHYRE models of plexin A1 (**C**) and NRP1 (**D**) show the predicted ribbon structures with corresponding cleavage sites denoted by color.

**Figure 3 ijms-22-03218-f003:**
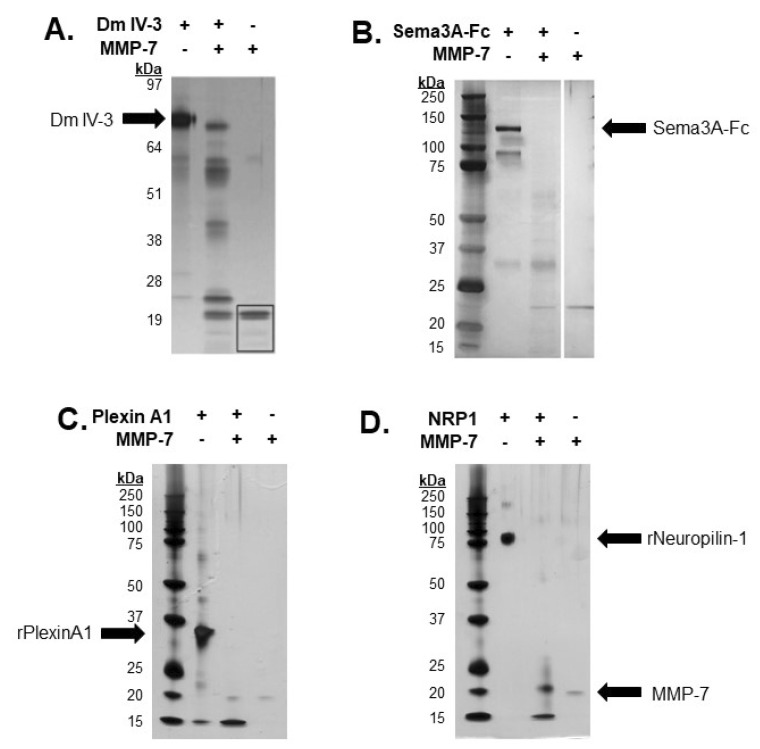
In vitro digestions of all PSPN Complex components demonstrate susceptibility to proteolysis by MMP-7. Silver stain of Dm IV-3 incubated with or without MMP-7 (adapted [5]) (**A**), MMP-7 digestion of Sema3A-Fc (0.75 µg) with or without MMP-7 (0.08 µg) overnight (adapted [4]) (**B**). Digestion of recombinant plexin A1 (2 µg) (**C**) and NRP1 (2 µg) (**D**) overnight with or without MMP-7 (0.2 µg). Black arrows indicate recombinant Dm IV-3, Sema3A-Fc, plexin A1, NRP1 and MMP-7.

**Figure 4 ijms-22-03218-f004:**
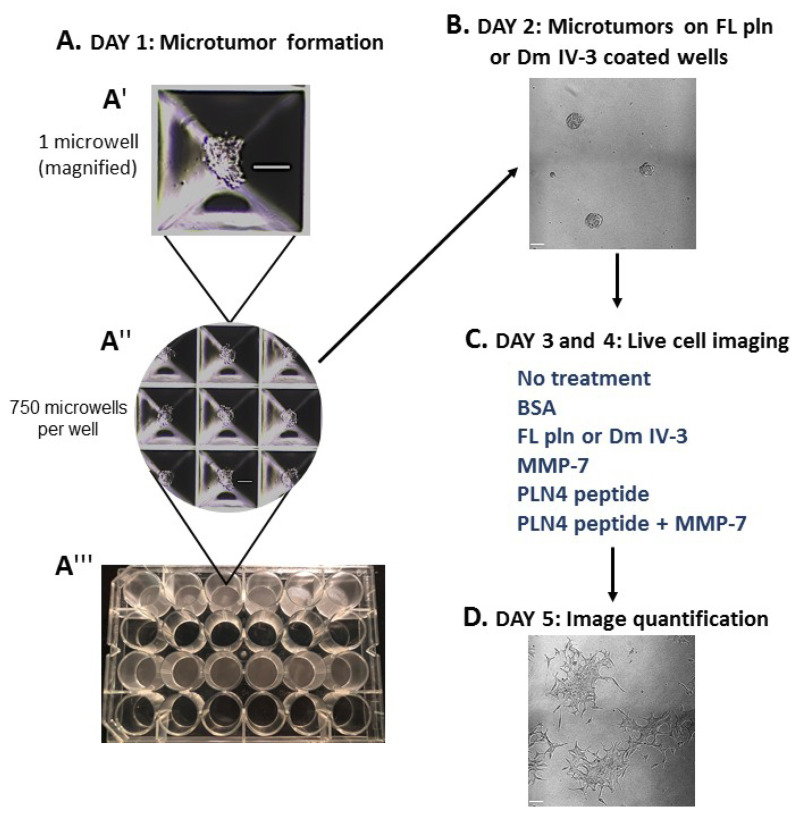
Microtumor formation assay. To evaluate the impact of FL pln, MMP-7 and PLN4 peptides on microtumor dispersion, uniformly sized C4-2 microtumors were pre-formed using a 24-well plate microwell system (Kugelmeiers AG. Zürich, Switzerland) (**A**). One single well of a 24-well plate contains 750 microwells that allowed us to form 750 microtumors of approximately 133 cells per microtumor (**A′**), (**A″**). These pre-formed microtumors were transferred to FL pln or Dm IV-3-coated wells for ~24 h (**B**). Next, microtumors were transferred again to uncoated wells or collagen I-coated wells, and the indicated consecutive treatments in (**C**) (blue text) were added to the cell culture media. Live-cell imaging (**D**) stopped at ~48 h and cell dispersion area was quantified (see Figure 5). Scale bars: 100 µm.

**Figure 5 ijms-22-03218-f005:**
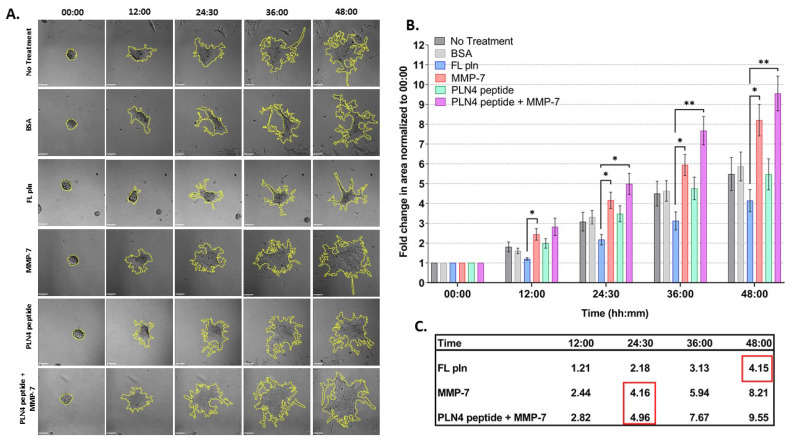
Time course analysis of microtumor dispersion under control conditions (no treatment, BSA) or in the presence of FL pln, MMP-7 and/or PLN4 peptide. (**A**) Representative micrographs showing cell dispersion of C4-2 microtumors with the indicated treatments and time points (scale bar: 100 µm). Yellow dotted lines on the micrographs show the region of interest corresponding to total microtumor dispersion area used to compute area fold change. (**B**) Evaluation of area fold change. Cell dispersion area is higher in microtumors treated with MMP-7 or MMP-7 in combination with PLN4 peptide compared to microtumors treated with FL pln (see text). Data are the mean ±SEM of three experimental repetitions, except two experimental repetitions for PLN 4 + MMP-7. * *p* < 0.05, ** *p* < 0.01. Two-way repeated-measures ANOVA with a *post hoc* Šídák test was performed. (**C**) Table shows mean values for fold change in area. Red wireframes demonstrate that the difference in time needed to reach a similar amount of dispersion in the different treatment conditions.

**Figure 6 ijms-22-03218-f006:**
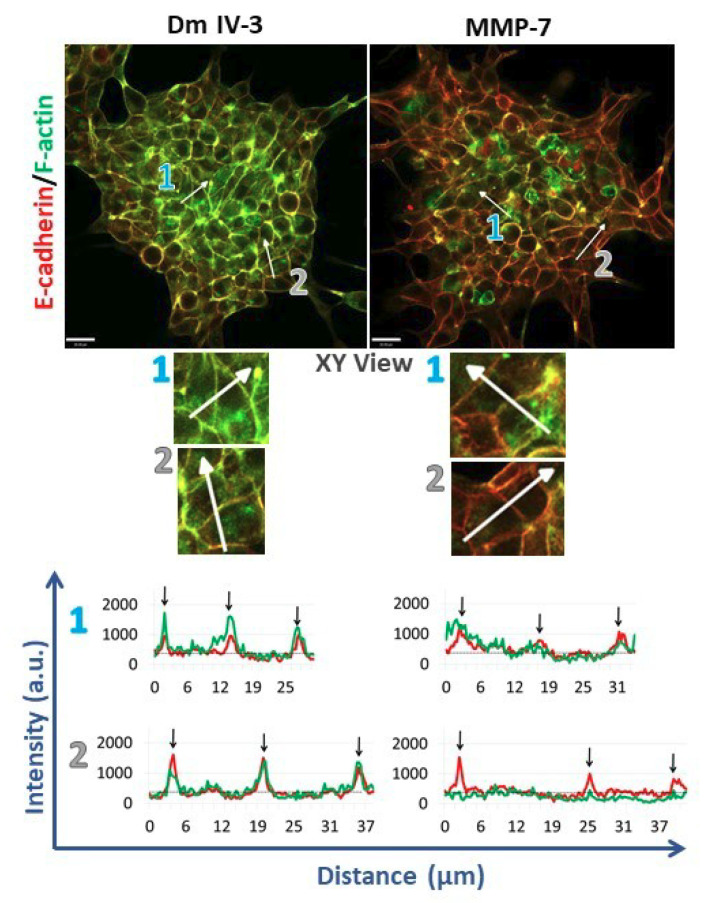
Immunostained microtumors for E-cadherin (red) and F-actin (green) at 24 h. Line scan analyses (see text) were performed for the regions highlighted by the white arrows. Arrow 1: microtumor center, arrow 2: microtumor periphery. Dm IV-3-treated microtumors show increase in co-aligned E-cadherin (red line) and F-actin (green line) at cell–cell contacts (black arrows) whereas microtumors treated with MMP-7 show a decrease in E-cadherin and F-actin co-alignment, indicative of loss of adhesion and initiation of cell dispersion. Line scans were performed on multiple cell–cell boundaries in microtumors under a variety of cohesive and dispersing conditions, then Pearson’s correlation analysis was performed on the same clusters at high magnification (3 clusters/condition). The images shown are representative of what we observed. Scale bars: 30 µm.

**Figure 7 ijms-22-03218-f007:**
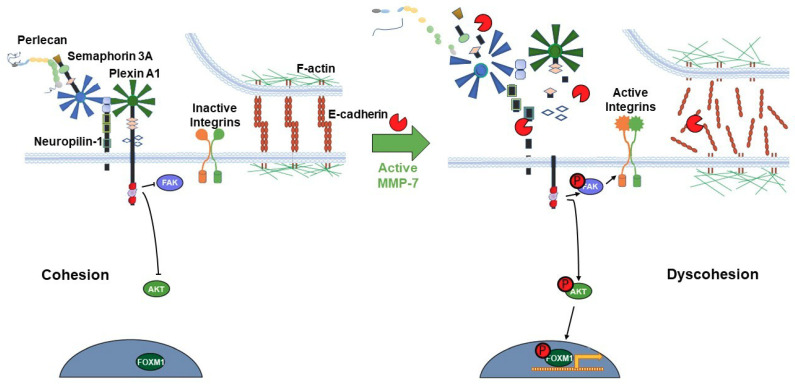
Proposed model of the PSPN Complex and E-cadherin/F-actin interaction in cohesive vs. dyscohesive PCa clusters. Model of the dynamic interactions between the PSPN Complex, MMP-7, and E-cadherin/F-actin. Perlecan and Sema3A secreted by the PCa surrounding stroma bind to and silence signaling from the plexin A1-NRP1 proteins. When unproteolyzed, this prevents activation of downstream FAK, AKT, and FOXM1, preventing dyscohesion. At this point, E-cadherin homodimers are intact and the cortical F-actin cytoskeleton is organized near the cell surface. Upon activation of MMP-7, cleavage of the PSPN Complex and extracellular E-cadherin ectodomain occurs, activating integrins, and releasing PCa cells to initiate dyscohesion and migration. Colors and shapes of the PSPN Complex components are described in Figure 1.

## Data Availability

Not applicable.

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
