# Peer review of "Cleavage of the Perlecan-Semaphorin 3A-Plexin A1-Neuropilin-1 (PSPN) Complex by Matrix Metalloproteinase 7/Matrilysin Triggers Prostate Cancer Cell Dyscohesion and Migration"

_ijms, 2021, doi:10.3390/ijms22063218_

Round 1
Reviewer 1 Report
Tellman et al.: Cleavage of the Perlecan-Semaphorin 3A-Plexin A1-Neuro-2 pilin-1 (PSPN) Complex by Matrix Metalloproteinase 7/Matri-3 lysin Triggers Prostate Cancer Cell Dyscohesion and Migration
This study found that MMP-7 contributes to prostate cancer cell dyscohesion and migration by cleavage of the PSPN complex.
This study relied on only one prostate cancer cell line C4-2. This affects the integrity of the results.
Western blot or IF to show the effect on the activation of downstream FAK, AKT, and FOXM1 should be included.
Author Response
In response to the suggestions from Reviewer 1, we have made the following changes:
- We performed a minor spell check and checked for abbreviation use to make consistent with journal standards.
- Regarding use of one cell line: We chose the C4-2 line among several others that we checked because it had two desirable properties for this study, namely that it responded to addition of perlecan and/or perlecan domain IV-3 by forming clusters very consistently, and second that it had low endogenous MMP-7 levels so that we could assess the actions of exogenous MMP-7. We saw similar behavior for the C4-2 parental cell line, LNCaP. An aggress subline of C4-2 overexpression RANKL that we described previously (PMID: 24478054) in which c-Met is upregulated also upregulated MMP-7 production that prevented the cells from clustering well, as expected from our model. Likewise, PC3 cells that expressed MMP-7 at easily detectable but lower levels than the LNCaP-RANKL cells formed only small clusters. We have added our cell line screening and observations to the new Supplemental Table (S1) and added this information to the manuscript including in Methods, Results and Discussion.
- We previously published the Western data showing the effect on activation of downstream FAK, AKT, and FOXM1 (PMID: 29740048) consequent to MMP-7 digestion that allowed us to include the downstream pathway in Figure 5. A copy of the figure from that paper is included in the attachment. At the time of that report, we were not yet aware of the entire composition of the PSPN complex. One of the things that we are excited about this report is that it let us bring a number of independent but related observations together to describe a new receptor complex controlling cohesion and dyscohesion and the downstream pathway to present a model for proteolytic control of metastatic behavior of prostate cancer cells.

Reviewer 2 Report
Minor changes: Please enter abbreviations in figure captions. Review the bibliography, for example reference 21, 33, 38 (include year in bold). Congratulations to the authors.Author Response
In response to Reviewer 2, we have entered abbreviations in the figure captions and reviewed the references for consistency, as suggested. We thank the reviewer for the enthusiasm for our work.

Round 2
Reviewer 1 Report
1. Figure 3. In vitro digestions of all PSPN Complex components demonstrates susceptibility to proteolysis by MMP-7. Silver stain of Dm IV-3 incubated with or without MMP-7 (adapted [5]) (A), MMP-7 digestion of Sema3A-Fc (0.75 μg) with 118 or without MMP-7 (0.08 μg) overnight (adapted [4]) (B).
-Figures 3A and B were from authors’ previous studies, should it consider to present in supplementary figure? Does any permission need to be get from re-using of these data?
2. We chose the C4-2 line among several others that we checked because it had two desirable properties for this study, namely that it responded to addition of perlecan and/or perlecan domain IV-3 by forming clusters very consistently, and second that it had low endogenous MMP-7 levels so that we could assess the actions of exogenous MMP-7. We saw similar behavior for the C4-2 parental cell line, LNCaP. An aggressive subline of C4-2 overexpressing RANKL that we described previously (PMID:24478054) in which c-Met is upregulated also upregulated MMP- 7 production that prevented the cells from clustering well, as expected from our model. Likewise, PC3 cells that expressed MMP-7 at easily detectable but lower levels than the LNCaP-RANKL cells formed only small clusters. We have added our cell line screening and observations to the new Supplemental Table and added this information to the manuscript including in Methods, Results and Discussion.
-Table S1 is not a new version.
-“We saw similar behavior for the C4-2 parental cell line, LNCaP.” The data for LNCaP must be added to burst up the quality of the paper. Data from one cell line is not appropriate standard for data reproductivity.
-These data supporting using C4-2 should be added to supplementary figures. Eg, LNCaP-RANKL cells formed only small clusters.
3. Figure 6: did the experiment performed n=3?
4. “We previously published the Western data showing the effect on activation of downstream FAK, AKT, and FOXM1 (PMID: 29740048) consequent to MMP-7 digestion that allowed us to include the downstream pathway in Figure 5.” As this downstream has been published previously. IF of the downstream pathway in Figure 6 showing the effect can confirm the link of the independent but related observations. This is the most important message of the study. Otherwise, it is just a descriptive linkage.
Author Response
We thank you for the speedy comments and are happy to provide the requested information. We believe there is sufficient new information here to warrant publication, and we do not believe this is a phenomenon attributed to one cell line, but rather to PCa subtypes with this phenotype: expression of SPN complex with low endogenous MMP-7.
Detailed responses below:
- “Figures 3A and B were from authors’ previous studies, should it consider to present in supplementary figure? Does any permission need to be get from re-using of these data?”
We wanted to show one combined figure illustrating that all four components of the PSPN Complex are digested with MMP-7, rather than piecemeal. We requested and obtained permission to adapt the previously published figures as presented, referring to the original. One was open source. We think it brings the story into focus to present this comprehensive figure.
- “Table S1 is not a new version.”
The table S1 (cell lines, PCS and MMP-7 ELISA) is entirely new, and the previous table S1 (dispersion assay repeated measures two way ANOVA) is now table S2. We uploaded this supplemental table and accompanying figure showing the clustering response of the various cell lines, but it did not appear for the reviewer for reasons we cannot explain. We hope this will successfully appear for the reviewer this time.
“These data supporting using C4-2 should be added to supplementary figures. Eg, LNCaP-RANKL cells formed only small clusters.”
The new table S1, shows the 10X images of clusters formed by the various lines in response to perlecan domain IV-3 and referred to in table S1.
- “Figure 6: did the experiment performed n=3?”
Multiple members of our laboratory, including three authors of this paper, have personally performed clustering and dyscohesion assays with perlecan/perlecan domain IV-3/MMP7, thus this aspect of data in Figure 6 has been performed at least dozens of times. It is routine for us to perform experiments in which measurements are taken with 3 technical and biological replicates. We also have immunostained tens if not hundreds of PCa microtumors with E-cadherin and actin, as it is a routine way to assess PCa cohesion in our labs. It was during this process that we noted that MMP-7 was “separating” the signal for E-cadherin and actin at the periphery of dissociating microtumors. The line scans were suggested to us as a way to quantitate the signal separation, which is what is shown in the bottom of Figure 6. Line scans were performed on multiple cell-cell boundaries in microtumors under a variety of cohesive and dispersing conditions, then Pearson’s correlation analysis was performed on the same clusters at high magnification (3 clusters/condition). The images shown are representative of what we observed.
- “As this downstream has been published previously. IF of the downstream pathway in Figure 6 showing the effect can confirm the link of the independent but related observations. This is the most important message of the study. Otherwise, it is just a descriptive linkage.”
We agree and note that the new work in this submission is part of an ongoing long term project in our lab in which each publication builds on the last as we learn more about this new role of perlecan in PCa progression. We independently validated this downstream pathway, initially identified in an RPPA experiment the complete results of which were published as supplementary material in PMID: 29740048. Building on the work presented in this submission, we showed by IF the suppression of active FOXM1 in the nucleus (figure below for the reviewers) by addition of DmIV-3 of perlecan. This work is the focus of a complete follow up manuscript focusing on FOXM1 activation and inactivation for which we are presently completing experiments. Including this here, even in a supplement, could compromise that publication and we respectfully request not to have to include it in this article in which we focus on MMP-7 cleavage of the PSPN complex. We note for the reviewer that this submission is the first to report MMP-7 cleavage of the entire PSPN complex associated with the activation of this pro-metastatic pathway, and that Figure 7 is the first to illustrate the pathway that we believe governs cohesion and dispersion of PCa cells by MMP-7 produced in the tumor microenvironment.
Figure for Reviewers can be found in the attached document.

Round 3
Reviewer 1 Report
- According to the reply from author dated “February 25, 2021” point #2, “We saw similar behavior for the C4-2 parental cell line, LNCaP.” It would be simply for author to show similar results in the supplementary data. Would this similar behavior only related to MMP-7 production and cluster formation ability, but not the microtumor dispersion ability and E-cadherin and F-actin rearrangement ?
- Figure 6, would author add the information to M&M/Figure legends “Line scans were performed on multiple cell-cell boundaries in microtumors under a variety of cohesive and dispersing conditions, then Pearson’s correlation analysis was performed on the same clusters at high magnification (3 clusters/condition). The images shown are representative of what we observed.”
Author Response
1. Representative images of the clustering data for LNCaP and C4-2 were included in the previous submitted Supplementary Table S1 (first revision). We have not systematically measured dispersion, E-cadherin or F-actin rearrangements in the LNCaP line, which we now note on Lines 263-264, but since both are PCS2 we would expect them to behave similarly. We chose the C4-2 line for this work over the parental LNCaP because our lab is interested in castrate resistant cell interactions in bone metastases.
Lines 152-155 We added text in the previous revision to clarify this point “C4-2 PCa cells were chosen among several PCa cell lines we tested (Table S1), because they responded to perlecan by clustering and did not produce detectable levels of endogenous MMP-7 that prevents clustering.”
We included a new Supplementary figure 1 based on analysis of the AKT pathway in 8 cell lines (see reply to Academic Editor). We also added our western data on active AKT in PCS2 and PCS3 representative lines.
2. The information was added to the figure.